SOFTWARE

# UNNT: A novel Utility for comparing Neural Net and Tree-based models

**Vineeth Gutta**[1]*, **Satish Ranganathan Ganakammal**[2], **Sara Jones**[2], **Matthew Beyers**[2], **Sunita Chandrasekaran**[1]

**1** Department of Computer & Information Sciences, University of Delaware, Newark, Delaware, United States of America, **2** Cancer Science Data Initiatives, Leidos Biomedical Research, Inc., Frederick National Laboratory for Cancer Research, Frederick, Maryland, United States of America

* vineethg@udel.edu

## Abstract

The use of deep learning (DL) is steadily gaining traction in scientific challenges such as cancer research. Advances in enhanced data generation, machine learning algorithms, and compute infrastructure have led to an acceleration in the use of deep learning in various domains of cancer research such as drug response problems. In our study, we explored tree-based models to improve the accuracy of a single drug response model and demonstrate that tree-based models such as XGBoost (eXtreme Gradient Boosting) have advantages over deep learning models, such as a convolutional neural network (CNN), for single drug response problems. However, comparing models is not a trivial task. To make training and comparing CNNs and XGBoost more accessible to users, we developed an open-source library called UNNT (A novel Utility for comparing Neural Net and Tree-based models). The case studies, in this manuscript, focus on cancer drug response datasets however the application can be used on datasets from other domains, such as chemistry.

**Data Availability Statement:** All relevant data are within the manuscript and its Supporting information files. The code and data are included in 'S1 Data' where the library and a smaller subset of

## Author summary

Advancement in data science, machine learning (ML), and artificial intelligence (AI) methods has enabled extraction of meaningful information from large and complex datasets that has assisted in better understanding, diagnosing, and treating cancer. The understanding of the drug response domain in cancer research has been accelerated with developing ML models to aid in predicting the effectiveness of the drugs based on a specific genomic molecular feature. In this study we developed a novel robust framework called UNNT (A novel Utility for comparing Neural Net and Tree-based models) that trains and compares deep learning method such as CNN and tree-based method such as XGBoost on the user input dataset. We applied this software to single drug response problem in cancer to identify the best performing ML method based on the National Cancer Institute 60 (NCI60) dataset. In addition, we studied the computational aspects of training each of these models where our results show that neither is evidently superior on both CPUs and GPUs while training. This shows that when both models

the data are also available on GitHub: https://github.com/vgutta/UNNT.git.

**Funding:** This study has been supported in part by the Joint Design of Advanced Computing Solutions for Cancer (JDACS4C) program established by the US Department of Energy (DOE) and the National Cancer Institute (NCI) of the National Institutes of Health (NIH) Leidos Biomedical Research contract no. 75N91019D00024. MB, SG, SJ acknowledge support from contract no. 75N9019D00024 received as employees of Frederick National Laboratory for Cancer Research. VVG acknowledges indirect financial support from contract no. 75N91019D00024 received through a subcontract with Frederick National Laboratory for Cancer Research. The data is hosted on the funding agency's infrastructure. Authors affiliated with FNL helped design the study, were integral for data selection, and also assisted in the analysis of the results. The decision to publish this study was a joint decision between the authors of this study and FNL.

**Competing interests:** The authors have declared that no competing interests exist.

have similar error rates for a dataset the hardware available determines the model choice for training.

## Introduction

To leverage machine learning (ML) for cancer applications, the National Cancer Institute (NCI) at the National Institutes of Health (NIH) in collaboration with the Department of Energy (DOE) established the Joint Design of Advanced Computing Solutions for Cancer (JDACS4C) program [1]. This program developed three pilot projects focused on cancer research: Pilot 1-cellular level; Pilot 2-molecular level; Pilot 3-population level [2]. Along with the pilots, NCI-DOE also developed the CANDLE (Cancer Distributed Learning Environment) [3] project for hyperparameter optimization (HPO) on the models.

Our work in this paper is related to a subset of the drug response problems addressed in Pilot 1-cellular level. Specifically, our work builds on the existing single drug response predictor models officially known as P1B3, which uses a deep neural network to model tumor growth based on gene expression, drug concentrations, and drug descriptors data. We compared the performances of the existing CNN-based P1B3, built within the CANDLE framework, with the new tree-based methods. We show that a tree-based method, like XGBoost, is a better model than neural network CNN when the training data, such as drug response data, is tabular.

### Background

Unlike computer vision models, which rely on unstructured data such as images, the CANDLE framework drug response, and other models, rely on structured data in tabular format. The big breakthrough in Deep Learning came because of the ability of neural networks to perform well on unstructured data such as images. Deep neural networks have been successfully adapted to various domains outside of computer vision such as natural language processing (NLP) [4] and with the CANDLE framework to various problems in cancer research [3]. We tested the existing CNN model architecture used in CANDLE's single drug response model with the NCI60 dataset and it peaked at an accuracy around 70%. Further improving such models will require data augmentation or changing to a new model architecture.

Recent studies [5] have shown that tabular data may not require complex black box models such as CNNs to perform well. Gradient boosted decision tree (GBDT) models such as XGBoost [6] can match or exceed the performance of deep learning models [5]. State-of-the-art deep learning models for tabular data perform worse than XGBoost when tested on new data not in their respective original studies [5]. In addition, more recent work investigates the differences between the models to help researchers understand the inductive biases of each type of model [7]. Another recent work conducts an in-depth survey comparing machine learning methods with deep learning approaches [8]. It also concludes that GBDT ensembles tend to outperform state-of-the art deep learning models for tabular datasets [8]. On the other hand, [9] takes a slightly different perspective compared to the previous two works cited [8] [7]. The authors of this work explore the properties of datasets that make them better suited for either Neural Networks or GBDTs. They find that GBDT models are better at handling skewed feature distributions compared to Neural networks. This is yet another study performed with the explicit goal of helping practitioners choose the best model for their work. All of these studies [8] [7] [9] help researchers and practitioners understand the strategies they should use for their own tabular datasets. In spite of all the analysis from previous works, there still remains a need for helping researchers and practitioners, with their own tabular datasets,

seamlessly compare the two model architectures rather than only rely on the analysis and comparisons using fixed datasets provided by recent studies. One of the major contributions of our work is to address that existing gap.

In the following two paragraphs, we summarize the major characteristics of the two types of models that our analysis in the rest of the paper is based on. Tree-based models are supervised learning methods that create decision trees based on the training data provided. Decision trees are nodes in the tree-based model that create a "split" at a particular point in the data range for a particular feature in the data inferring rules during this process. This model uses the Exact Greedy Algorithm [6] for split finding. They can be used for both classification and regression problems. The main difference between them when splitting occurs due to differing metrics used to minimize loss. And unlike decision trees, regression trees contain a score on each leaf that is used to calculate the final score by summing the leafs [6]. They are formally known as Gradient Boosted Regression Trees (GBRT) and are part of a larger class of method known as Classification and Regression Trees (CART). During training, XGBoost builds decision trees sequentially and then uses a technique known as boosting where each successive tree gives more weight to examples that were previously misclassified. After each iteration, XGBoost computes the gradients of loss functions based on predictions and then creates a new decision tree to reduce errors made by previous trees [6].

The CNN model consists of a multilayer perceptron (MLP) and is a feed forward neural network. It generally consists of an input layer, hidden layers, and an output layer. The input layer receives the data and the hidden layers learn a continuous function based on the training data. These consist of convolutional layers called filters (kernels) that slide over the input data and compute the dot product between their weights and the input data by computing the summation of the product of corresponding elements in two matrices [10]. Following this operation, known as a convolution, non-linearity is introduced into the network using an activation function to learn complex relationships between features. Then a pooling layer, also using a kernel, slides across the data to reduce spatial dimensions and overfitting [11]. A series of convolutional and pooling layers are typically followed by at least one fully connected layer to learn high-level features extracted by previous layers and the relationships between those features. After the forward pass through the network, a loss function is used to compare CNN's output to the ground truth and is used to update the network's weights and biases using backpropagation and gradient descent. Backpropagation computes the gradient of the loss functions for each of the weights and biases in the network going back to the input layer and these gradients are used to update the model parameters to minimize loss using optimization techniques such as gradient descent [10]. Finally the output layers perform predictions which can be classification or a numerical output in a regression problem. For classification, a softmax layer converts the raw output from the network into class probabilities. Many other regularization techniques such as dropout are applied to prevent overfitting and improve model performance [11]. This network architecture achieved state-of-the-art results in domains with grid-like unstructured data such as images but is not ideal for structured data.

## Design and implementation

Recent works such as [8], [7], and [9], which are complementary to our work, explain how various models and datasets they explore perform with tabular datasets. The extensive analysis in these papers [8] [7] [9] still leaves a gap where domain scientists and researchers need to validate and test with their own data the findings of the recent works cited. Their work further reinforces the need for an abstraction that allows users to quickly test their own data and see if

the conclusions in the studies can be replicated. We address a major gap that remains because a user must eventually apply the findings in these studies to their own datasets.

To address this gap and provide users with structured (tabular) data the option to compare both the CNN and XGBoost models, we developed an open-source comparative library called UNNT that allows users to bring their data to train both models. See Fig 1 for a flowchart that represents the library in S1 Fig. The XGBoost model relies on the open source libraries from Distributed (Deep) Machine Learning Community (DMLC) which UNNT uses to provide users the ability to train XGBoost models [6].

For data preprocessing, calculating metrics after training, and displaying results, we rely on other packages such as Pandas, Scikit-Learn, Numpy, Matplotlib. To build CNN models we employ some of the functionality provided by the CANDLE library in Pilot 1 Benchmark 3 to do data preprocessing, model definition and instantiation, and model training. This has dependencies such as TensorFlow 1.0. Also, we use scikit-learn to import metrics to quantify model performance.

The preprocessing steps are important for the predictive accuracy of our pretrained models to work on any data users bring to train CNN and XGBoost models. Because the exact format

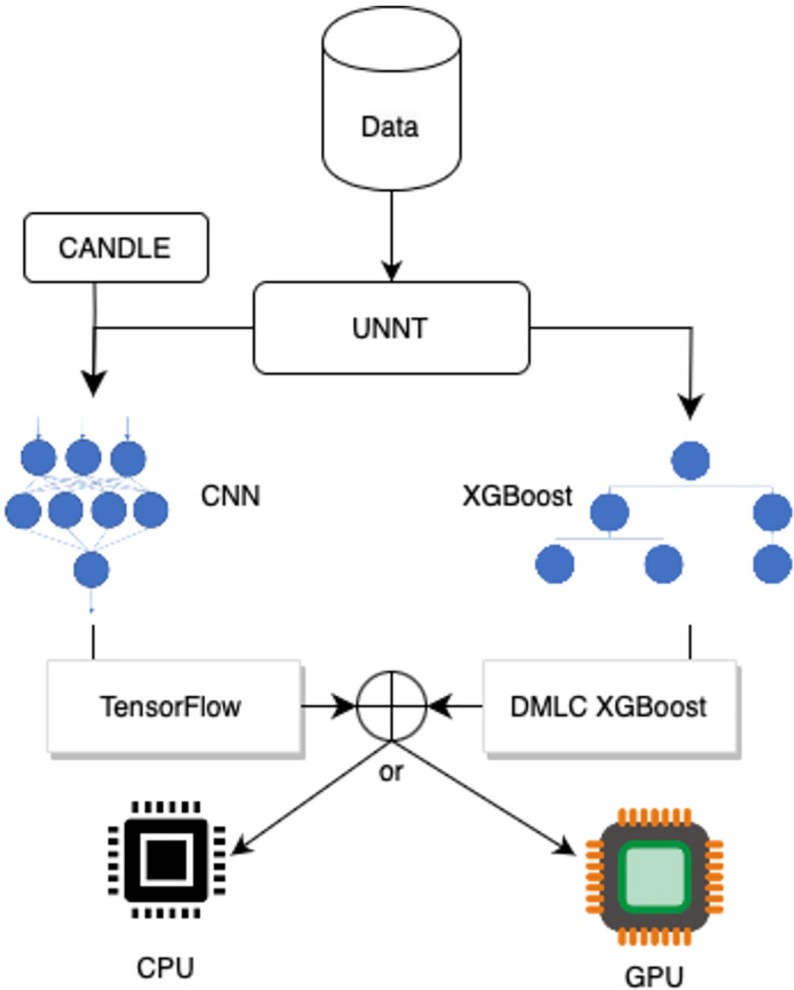

**Fig 1. Flow chart of UNNT.** Depicts how UNNT works by combining existing libraries and adding functionality to create multiple models with the same dataset with the option to run on a CPU or a GPU.

of user provided datasets is unknown, users are responsible for any preprocessing and data cleaning steps that would be necessary prior to model training. Users need to decide what features they want to keep in the dataset being used to train and test the models and our models should be able to accommodate data of any shape as long as it fits into the device memory. If the data does not fit into the device memory of the system being used if will require the user to distribute the data and compute across more than one CPU/GPU. Libraries such as cuDF [12] can be used to distribute data across NVIDIA GPUs and Dask [13] can be used to distribute compute across CPUs/GPUs.

Once users provide their data, UNNT splits that data into training, validation, and testing sets and the user can specify the percentage of data used for testing and validation. The defaults are 70% training and 30% test data for XGBoost while half the test data is used for validation of CNN model. We use scikit-learn's train_test_split() method to randomly sample since we are comparing it to a CNN model that randomly sampled. We convert each of the data splits into numpy arrays to train using both CNN and XGBoost. There are several hyperparameters that can be set, for both CNN and XGBoost models, and we will have recommended default parameters which can be customized by the user. It's important to note that compared to XGBoost CNN models are more complex and have more hyperparameters. In many cases there is little overlap between them. Users have full control of the parameters tested to find the best combination of hyperparameters for their dataset for each of the models.

Finally UNNT provides users error metrics to evaluate both models trained using their data including $R^2$ and Root mean square error (RMSE).

## Data

The data for the study was obtained from The Predictive Oncology Model and Data Clearinghouse (MoDaC) [14] data warehouse that was released as part of JDACS4C [1]. The dataset includes RNA-Seq expression profiles, drug response and molecular drug descriptors for National Cancer Institute 60 (NCI60) [15] cell lines.

The RNA-Seq expression data from these various cell lines are normalized using ComBat-seq [16]. Drug dose response data obtained from these cell lines are normalized using a Hill slope model with three bounded parameters [17] and drug molecular descriptors were generated using Dragon software package (version 7.0) [18].

NCI60 data includes a combination of gene expression values [19], drug response values [20], and drug descriptors [21] found in MoDaC [14].

Our NCI60 data consists of data similar to the NCI60 data the original Single Drug Response model in CANDLE Benchmarks was trained with, with two main differences. Firstly our dataset uses only using lincs1000 genes [22] in the RNA sequence gene expression data instead of all the protein coding genes, due to the importance of those genes in the dataset. Secondly, the original NCI60 dataset in drug response data uses 'Growth' and 'Concentration' values where 'Growth' is the target variable and 'Concentration' is a feature which we no longer use. 'Growth' is replaced with 'AUC' data in the drug response dataset because area under the curve (AUC) [23] is a more definitive parameter that combines both potency (concentration) and efficacy of a drug and is more robust when comparing a single drug across multiple cell lines for similar dose levels.

## Results

### Experimental setup

The compute resources for this work used NSF sponsored cluster, DARWIN, [24] at UDEL and Perlmutter [25], at LBNL. DARWIN and Perlmutter have GPU and CPU nodes.

DARWIN uses NVIDIA V100 GPUs while Perlmutter has the latest A100 GPUs. DARWIN has AMD EPYC 32 core processors which are similar to Perlmutter that has an AMD EPYC 7713 64-core CPU.

## XGBoost

We trained an XGBoost model using NCI60 expression, dose response, and drug descriptor data with AUC as the target (predictor) variable. Our observed test accuracy yielded 0.83 for $R^2$ score and 0.05 RMSE score. In addition, we created a separate dataset set aside before training and included 10% of the cell lines. Table 1 shows the difference in test errors between the two.

To find the best set of hyperparameters, we performed hyperparameter optimization using grid search technique and cross validation. Grid search trains a new model for every combination of hyperparameters while cross validation uses a different subset as test data to get an average across five subsets. Best set of hyperparameters found were ETA:0.1, Max depth: 10, Subsample: 0.5, N estimators:500. We used these hyperparameters to train a new model and the results are shown in Table 1. Hyperparameter optimization led to slight improvement that was lower than our initial expectations. This was a result of well documented ranges for the various parameters and thus we happened to choose values that were close to optimal for each of the hyperparameters.

XGBoost model training requires the training data to be fully merged before training commences and this resulted in memory issues. To solve the memory issues, we experimented with smaller datasets and reduced the number of drugs from 30,000 to 159, based on an FDA approved drugs list [20]. Results found in Table 2. The list can be found in the file S1 FDA.

## CNN

We trained the original CNN model using the new NCI60 data as described in the data section above. The results shown in Table 3 were after applying hyperparameter optimization (HPO) and to determine the best parameters where the learning rate is 0.01 and tanh as the activation function. The model failed to converge on other activation functions such as ReLU for NCI60 data.

**Table 1. XGBoost Errors for model trained NCI60 data.**

|  | $R^2$ score | RMSE score |
|---|---|---|
| **Test error** | 0.82 | 0.051 |
| **Test error with new cell lines** | 0.76 | 0.065 |

**Table 2. XGBoost Errors for model trained on NCI60 data with only FDA approved drugs list.**

|  | $R^2$ score | RMSE score |
|---|---|---|
| **Test error** | 0.84 | 0.069 |
| **Test error with new cell lines** | 0.66 | 0.094 |

**Table 3. CNN errors for NCI60 after training after performing HPO.**

|  | $R^2$ score | RMSE score |
|---|---|---|
| **Test error** | -30.32 | 0.81 |

As shown in in Table 3, the CNN model performed much worse than XGBoost when trained on NCI60 data. The RMSE metric's best possible value is 0 and it can go to infinity, but a value such as 0.81 does not give us good insight into the quality of the regression model. On the other hand, the negative $R^2$ value for the test score indicates the model performance is poor but its magnitude does not indicate how poorly it performed. The best value for $R^2$ is 1, meaning the model completely explains predicted data variability [26]. These results indicate that both $R^2$ and RMSE scores show XGBoost model is outperforming the CNN model trained on this NCI60 dataset. It is important to emphasize that the choice of $R^2$ was highly dependent on the nature of the dataset used in this study and may not be widely applicable. For more analysis of the advantages of $R^2$ score for datasets similar to what we used for this study see [26].

## Training times of CNN and XGBoost

The original CNN model is capable of running on both GPUs and CPUs, by design, since it is built with the TensorFlow framework. While the XGBoost model runs on CPU by default it can also be trained on GPU where we use the parameter *tree_method="gpu_hist"* in the XGBRegressor function. This means both models in UNNT can be accelerated using GPUs. In the following section we show comparisons of model training times with CPUs and GPUs. Comparison of CPU threads vs single GPU performance shown in Table 4.

**Training of full NCI60 drugs.** In addition to the model built using only the FDA approved drugs, we also built an XGBoost model using all the available drug data we have access to, however, use of 30,000 drugs presented many challenges due to the volume. The main challenge using the entire drug list entailed finding a system with at least 500GB of memory for the merged data before we train the XGBoost model.

CNN models can have varying training times based on parameters used for training such as subsampling with fewer features, specifying fewer training, validation, test steps, and by reducing the number of training epochs. We discuss some of those results. Table 5 shows that the

**Table 4. Results using XGBoost.** Times for threads represents model runs on CPU with the corresponding threads used for speedup. Last row corresponds to running the same model on single NVIDIA V100 GPU.

| Threads | Time (hours) |
|---|---|
| 1 | 5.6 |
| 2 | 6 |
| 4 | 5.94 |
| 8 | 6 |
| 16 | 5.9 |
| 32 | 6.7 |
| 64 | 7.2 |
| V100 GPU | 0.3 |

**Table 5. CNN model trained with all features on an NVIDIA V100 GPU.**

| Epochs | $R^2$ score | RMSE score | Time (hours) |
|---|---|---|---|
| 1 | -30.46 | 0.821 | 2.3 |
| 5 | -30.32 | 0.819 | 11.4 |
| 10 | -30.32 | 0.819 | 22.96 |
| 15 | -30.32 | 0.819 | 34.36 |

**Table 6. CNN model trained with all features on a CPU.**

| Epochs | $R^2$ score | RMSE score | Time (hours) |
|---|---|---|---|
| 1 | -30.35 | 0.81 | 1.1 |
| 5 | -30.31 | 0.81 | 5 |
| 10 | -30.32 | 0.81 | 9.97 |
| 15 | -30.32 | 0.81 | 15 |

**Table 7. Fastest training times for CNN and XGBoost on CPU and GPU (all features).**

|  | CPU (hours) | GPU (hours) |
|---|---|---|
| **CNN** | 1.1 | 2.3 |
| **XGBoost** | 5.2 | 0.3 |

CNN model does not improve as the number of epochs increases, eliminating a benefit of higher epochs. And the difference between training times of CNN and XGBoost is large. Table 6 shows that CNN model converges to its optimal learning capacity in 1 epoch, hence it would still take three times longer to train than an XGBoost model trained with a V100 GPU in the best case scenario of 1 epoch.

When comparing training times from Tables 5 and 6 we can see that the CNN model takes half as long to train on CPU compared with training on GPU. This is most likely a result of the size of the dataset where data transfer from CPU to GPU becomes a bottleneck and increases training time.

**Training on FDA approved drugs.** Table 7 shows us that an XGBoost model trains much faster on a GPU when training on a dataset that only contains FDA drug subset. We observed that the training times on the CPU increased as more threads were added. This occurs when the communication overhead is greater than the computational benefit of distributing a model across cores.

Table 8 shows the one instance where CNN model trains faster than an XGBoost model when training on a similar dataset. Comparing Tables 8 and 9 we see the CNN model trains faster on a CPU even with less training data. The CPU results for CNN are not broken down by the number of threads because TensorFlow 1.0, the framework used to build CNN model, does not support threading on CPUs.

**Table 8. Results using XGBoost.** Times for threads represents model runs on CPU with the corresponding threads used for speedup. Last row corresponds to running the same model on single NVIDIA V100 GPU.

| Threads | Time (seconds) |
|---|---|
| 1 | 1495.44 |
| 2 | 1527.33 |
| 4 | 1534.78 |
| 8 | 1549.24 |
| 16 | 1568.54 |
| 32 | 1599.01 |
| 64 | 1639.83 |
| V100 GPU | 160 |

**Table 9. CNN model FDA drugs trained on a CPU.**

| Epochs | $R^2$ score | RMSE score | Time (seconds) |
|---|---|---|---|
| 1 | -29.76 | 0.81 | 130 |
| 5 | -30.18 | 0.81 | 592 |
| 10 | -30.45 | 0.81 | 1186 |
| 15 | -31.02 | 0.82 | 2355 |

**Table 10. CNN model FDA drugs trained on a V100 GPU.**

| Epochs | $R^2$ score | RMSE score | Time (seconds) |
|---|---|---|---|
| 1 | -30.07 | 0.81 | 300 |
| 5 | -30.96 | 0.81 | 1356 |
| 10 | -30.33 | 0.81 | 2672 |
| 15 | -31.35 | 0.81 | 5348 |

**Table 11. Fastest training times for CNN and XGBoost on CPU and GPU (FDA model).**

| | CPU (seconds) | GPU (seconds) |
|---|---|---|
| **CNN** | 592s | 1356s |
| **XGBoost** | 1495s | 160s |

Tables 7 and 10 show the advantage of training XGBoost on a V100 GPU that is consistently the fastest for the same data. Finally, Table 11 validates that using a GPU for XGBoost is optimal for training even with a smaller dataset.

## Conclusions

Exploring a niche domain such as drug response modeling for cancer cell lines, we show that utilizing a neural network (CNN) does not yield the best results. Instead, we show the impact of tree-based XGBoost model over a CNN model especially when the datasets trained on are tabular and running on a GPU. Our results demonstrate that using the same dataset, an XBoost model is faster than a CNN model while running on an NVIDIA GPU. An observable downside to using XGBoost is the larger memory requirement for training, as documented in this work, and this varies depending on the size of the dataset. As part of this work, we have developed a software, UNNT, that allows users to bring their own data and build models such as CNNs and XGBoost as well as compare how the models perform on their dataset. Thus UNNT makes a useful software for domain scientists to experiment with two unique model architectures for tabular data.

## Supporting information

**S1 Text. UNNT installation + running instructions.**
(PDF)

**S1 FDA. FDA approved drug list.**
(PDF)

**S1 Data. unnt.zipfile with the software and data.**
(ZIP)

**S1 Fig. Flowchart of UNNT.**
(TIFF)

## Author Contributions

**Conceptualization:** Vineeth Gutta, Satish Ranganathan Ganakammal, Matthew Beyers, Sunita Chandrasekaran.

**Data curation:** Vineeth Gutta, Satish Ranganathan Ganakammal, Sara Jones.

**Formal analysis:** Vineeth Gutta.

**Funding acquisition:** Sunita Chandrasekaran.

**Investigation:** Vineeth Gutta.

**Methodology:** Vineeth Gutta, Satish Ranganathan Ganakammal.

**Project administration:** Matthew Beyers, Sunita Chandrasekaran.

**Resources:** Sunita Chandrasekaran.

**Software:** Vineeth Gutta.

**Supervision:** Satish Ranganathan Ganakammal, Sara Jones, Matthew Beyers, Sunita Chandrasekaran.

**Validation:** Vineeth Gutta.

**Visualization:** Vineeth Gutta.

**Writing – original draft:** Vineeth Gutta.

**Writing – review & editing:** Vineeth Gutta, Satish Ranganathan Ganakammal, Sara Jones, Matthew Beyers, Sunita Chandrasekaran.

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
