## [Decision Letter · Decision Letter 0]

12 Jan 2024

Dear Mr. Gutta,

Thank you very much for submitting your manuscript "UNNT: A novel Utility for comparing Neural Net and Tree-based models" for consideration at PLOS Computational Biology.

As with all papers reviewed by the journal, your manuscript was reviewed by members of the editorial board and by several independent reviewers. In light of the reviews (below this email), we would like to invite the resubmission of a significantly-revised version that takes into account the reviewers' comments.

We cannot make any decision about publication until we have seen the revised manuscript and your response to the reviewers' comments. Your revised manuscript is also likely to be sent to reviewers for further evaluation.

Sincerely,

Samuel V. Scarpino

Academic Editor

PLOS Computational Biology

Mark Alber

Section Editor

PLOS Computational Biology

Reviewer's Responses to Questions

**Comments to the Authors:**

Reviewer #1: This paper prsents a comparison of CNN with XBoost algorithms in the decision of cancer drug responses. It is a nice application and the authors can improve their research as future work. The authors informed the reader that the suggested methods are applicable in the field of chemistry as well. The authors should give information about the research such that they will appliy the algorithms in classification or prediction purpose and give necessary references in the suggested algorithms with their applications.

Reviewer #2: Dear Authors,

The manuscript "UNNT: A novel Utility for comparing Neural Net and Tree-based models" presents a comparison between a decision tree-oriented method and deep learning networks in the medical field. The topic is interesting given the profusion of deep learning techniques in scientific works in different areas of application.

Regarding the article, check the punctuation, especially the use of commas. When an acronym is inserted for the first time in the text, its meaning should be presented. Pay attention to long paragraphs as they tend to make reading confusing. I have marked in the comments of the digital file the passages where the writing should be improved. Please revise and delete repetitive information from the text.

I would like your attention to the following recommendations:

1) Lines 59 to 81: cite the bibliographical references that provide support for the information presented.

2) Line 102: "UNNT splits that data into training, validation, and testing sets" present the percentage for training, validation and testing. Don't forget to detail the method used to draw the samples.

3) Line 105: "can be set, for both CNN and XGBoost models" . At this point, please note that the configuration of the convulational network is much more complex than the Random Forest or CARTO methods;

4) Please detail which type of decision tree associated with the XGBoost method was used in the study.

5) Is it possible to run the solution developed in the Cloud? It would also be interesting to point out that a very specific hardware resource was used, which may be beyond the reach of many researchers.

6) Line 135: add as supplementary information at the end of the article.

7) Cite in the text the flowchart of the steps implemented, which is presented at the end of the article.

8) Results: it would be interesting to also present the overall accuracy and precision values. To diversify the presentation of the results, it would be interesting to present graphs, especially the AUC.

9) Still on the results obtained, it would be interesting to compare them with the results obtained by the authors cited in the references.

10) Line 196: What is the criterion for defining a sample of 0.5% of the sample set? Please detail what is specified in the FDA approved list.

I end my review by congratulating you on your study.

Respectfully,

**Have the authors made all data and (if applicable) computational code underlying the findings in their manuscript fully available?**

Reviewer #1: Yes

Reviewer #2: Yes

PLOS authors have the option to publish the peer review history of their article (what does this mean?). If published, this will include your full peer review and any attached files.

Reviewer #1: No

Reviewer #2: **Yes: **MARCOS BENEDITO SCHIMALSKI
---

## [Decision Letter · Decision Letter 1]

11 Mar 2024

Dear Mr. Gutta,

Thank you very much for submitting your manuscript "UNNT: A novel Utility for comparing Neural Net and Tree-based models" for consideration at PLOS Computational Biology. As with all papers reviewed by the journal, your manuscript was reviewed by members of the editorial board and by several independent reviewers. The reviewers appreciated the attention to an important topic. Based on the reviews, we are likely to accept this manuscript for publication, providing that you modify the manuscript according to the review recommendations.

I agree with R3 that a bit more is needed in terms of literature review in the intro/discussion and an expanded discussion around the interpretation, etc. I don't believe additional analyses are needed, but I strongly encourage the authors to take R3's comments seriously.

Sincerely,

Samuel V. Scarpino

Academic Editor

PLOS Computational Biology

Mark Alber

Section Editor

PLOS Computational Biology

I agree with R3 that a bit more is needed in terms of literature review in the intro/discussion and an expanded discussion around the interpretation, etc. I don't believe additional analyses are needed, but I strongly encourage the authors to take R3's comments seriously.

Reviewer's Responses to Questions

**Comments to the Authors:**

Reviewer #1: The necessary updates has been given.

Reviewer #2: Dear Authors,

The manuscript "UNNT: A novel Utility for comparing Neural Net and Tree-based models" in its second version, has several changes that have made reading even more fluid. I have carried out a thorough reading and your manuscript is now ready for publication.

Congratulations,

Reviewer #3: This manuscript compares the performance of a convolutional neural network (CNN) and a tree-based method, XGBoost, on a tabular drug response dataset using the software library UNNT developed by the authors.

The limitations of deep learning methods for tabular data and the rigorous comparison of machine learning methods are important topics. The author’s work forms a solid foundation for contributing to these areas but would need further development in several areas to make it an improvement beyond existing techniques. I encourage the authors to continue to improve their work in the areas listed below.

Major Issues:

On the machine learning side, the literature review is of limited scope and the references are rather dated. As mentioned above, the difficulties of deep learning on tabular data are an active area of research and the authors need to engage more with recent publications on the subject. Reference [5] in the manuscript is a good starting point, another one could be Grinsztajn, Oyallon, & Varoquaux (NeurIPS 2022).

The authors frame UNNT as a general tool for the comparison of deep learning models and tree-based models on tabular data but in its current form only two models, CNN and XGBoost, are available. Supporting a greater variety of models, for example deep learning methods specifically designed for tabular data (see e.g. the review of Borisov et al. (IEEE TNNLS 2022)), would make UNNT more useful for other researchers. The same applies to the availability of evaluation metrics.

Related to the two comments above, I encourage the authors to state more clearly where they see the added value of using UNNT instead of working directly with the machine learning frameworks it is built on. These frameworks already provide many of the features the authors list under Future Directions in their manuscript. It is thus necessary to thoroughly justify why UNNT is needed.

As much of the model evaluation is based on the R2 score, a more thorough discussion of the applicability of this score to non-linear regression models is required. The limitations of R2 for non-linear models should be discussed openly. This includes the difference in interpretation in the linear and non-linear case. I would recommend not to use “R2 error” but rather “R2 score” or simply “R2”.

**Have the authors made all data and (if applicable) computational code underlying the findings in their manuscript fully available?**

Reviewer #1: Yes

Reviewer #2: Yes

Reviewer #3: Yes

PLOS authors have the option to publish the peer review history of their article (what does this mean?). If published, this will include your full peer review and any attached files.

Reviewer #1: No

Reviewer #2: No

Reviewer #3: No

Figure Files:

Data Requirements:

Reproducibility:

References:

---

## [Editor Report · Decision Letter 2]

28 Mar 2024

Dear Mr. Gutta,

We are pleased to inform you that your manuscript 'UNNT: A novel Utility for comparing Neural Net and Tree-based models' has been provisionally accepted for publication in PLOS Computational Biology.

Best regards,

Samuel V. Scarpino

Academic Editor

PLOS Computational Biology

Mark Alber

Section Editor

PLOS Computational Biology

---

## [Editor Report · Acceptance letter]

25 Apr 2024

PCOMPBIOL-D-23-01452R2 

UNNT: A novel Utility for comparing Neural Net and Tree-based models

Dear Dr Gutta,

I am pleased to inform you that your manuscript has been formally accepted for publication in PLOS Computational Biology. Your manuscript is now with our production department and you will be notified of the publication date in due course.

With kind regards,

Anita Estes
